# Deep learning analysis of fMRI data for predicting Alzheimer's Disease: A focus on convolutional neural networks and model interpretability

Xiao Zhou[2], Sanchita Kedia[3], Ran Meng[2], Mark Gerstein[1,2,3,4,5] *

1 Program in Computational Biology & Bioinformatics, Yale University, New Haven, CT, United States of America, 2 Department of Molecular Biophysics & Biochemistry, Yale University, New Haven, CT, United States of America, 3 Department of Computer Science, Yale University, New Haven, CT, United States of America, 4 Department of Statistics & Data Science, Yale University, New Haven, CT, United States of America, 5 Department of Biomedical Informatics & Data Science, Yale University, New Haven, CT, United States of America

* mark@gersteinlab.org

**Data Availability Statement:** The data underlying the results presented in this study are available from the Alzheimer's Disease Neuroimaging

## Abstract

The early detection of Alzheimer's Disease (AD) is thought to be important for effective intervention and management. Here, we explore deep learning methods for the early detection of AD. We consider both genetic risk factors and functional magnetic resonance imaging (fMRI) data. However, we found that the genetic factors do not notably enhance the AD prediction by imaging. Thus, we focus on building an effective imaging-only model. In particular, we utilize data from the Alzheimer's Disease Neuroimaging Initiative (ADNI), employing a 3D Convolutional Neural Network (CNN) to analyze fMRI scans. Despite the limitations posed by our dataset (small size and imbalanced nature), our CNN model demonstrates accuracy levels reaching 92.8% and an ROC of 0.95. Our research highlights the complexities inherent in integrating multimodal medical datasets. It also demonstrates the potential of deep learning in medical imaging for AD prediction.

## Introduction

Alzheimer's disease (AD) is a progressive neurological disorder that contributes significantly to mortality and morbidity, making it one of the leading causes of death in the United States [1]. This disease is primarily characterized by the accumulation of amyloid-beta plaques and tau tangles in the brain, leading to synaptic dysfunction and neuronal loss. As a result, AD typically begins with minor memory loss and gradually progresses to a more severe loss of the ability to interact with the environment. Recent studies indicate that this disease generally results in death, as no definitive cure is currently available [2, 3]. Although a cure remains elusive, early detection of AD could offer significant benefits, allowing for the early application of therapeutic interventions to preserve critical functions and potentially slow the disease's progression using advanced diagnostic tools like functional Magnetic Resonance Imaging (fMRI)

Initiative (ADNI), a third-party source. Data cannot be shared publicly due to ethical and legal restrictions as they contain sensitive patient information, the restrictions are imposed by ADNI Data Use and Publications Committee. However, researchers can apply for access through the ADNI database to replicate or extend the findings of this study. Access details are available at https://adni.loni.usc.edu/data-samples/adni-data/#AccessData. For administrative inquires, researchers may contact via ida@loni.usc.edu. The authors have obtained permission to use the ADNI data set in accordance with ADNI's Data Use Agreement. Other researchers should be able to access the data in the same manner by completing the application process available at https://adni.loni.usc.edu/data-samples/adni-data/#AccessData.

**Funding:** The author(s) received no specific funding for this work.

**Competing interests:** NO authors have competing interests.

analyzed through deep learning techniques such as Convolutional Neural Networks. This highlights the importance of innovative research in neuroimaging for improving the quality of life for affected individuals [4].

The prediction of AD is a significant challenge due to the complexity of its pathogenesis, which is believed to involve a combination of age-related brain changes, genetic mutations, and various environmental and lifestyle factors [5, 6]. Consequently, research has dived into different aspect based on available data sources. For instance, neuroimaging-related researches utilizes sophisticated imaging techniques such as magnetic resonance imaging (MRI), functional MRI (fMRI), structural MRI (sMRI), positron emission tomography (PET), and diffusion tensor imaging (DTI) to capture detailed information from brain structure, functions, and status for the analysis and prediction of AD [7]. For example, Jagust et al. found that changes in regions like the angular gyrus, mid-temporal gyrus, and hippocampus have been linked to AD in various studies [8]. In [9], the researches specifically focused on fMRI for brain analysis due to its advantage in detecting functional changes in the brain—which often precede of the structural changes. From a distinct perspective, genetics attribute significantly to Alzheimer's Disease (AD), with a substantial portion of cases arising from genetic mutations. For instance, missense mutations in the S182 gene was proved to be associated with the AD3 subtype of early-onset familial Alzheimer's disease (AD) [10]. Acker et al. [11] explores gene risk factors for Alzheimer's Disease (AD), which differ from direct genetic mutations as it tends to causing non-early-onset familial AD. The focus of their work is on genetic variations that increase susceptibility to AD that do not directly cause the early-onset of familial AD, such as the roles of APOE4 [12], BIN1, CD2AP, PICALM, PLD3, and TREM2. Furthermore, in [13], the authors discussed replicated loci in hitherto uncharacterized genomic intervals on chromosomes that may be related to AD, besides from CLU, PICALM, etc.

Since its introduction, deep learning and machine learning is gaining increased popularity as a novel approach on AD research [14, 15], on both neuroimaging and gene perspective. For example, Sethuraman et al. developed custom deep learning models, including a modified AlexNet and Inception V2, for Alzheimer's Disease prediction using resting-state fMRI data [16]. The standout result was from their Ensemble Deep Learning Model (D2), particularly with the slow5 frequency band, which achieved an accuracy of 96.61% and an AUC of 0.9663, demonstrating the potential of deep learning in neuroimaging for accurate AD diagnosis. Additionally, in [17], the paper presents a classification algorithm for Alzheimer's Disease (AD) using dynamic functional connectivity patterns in resting-state fMRI. The model employs maximum value of wavelet coherence fluctuations (MWCF) and analysis of variance (ANOVA) for feature selection. The best performance was achieved using a cubic SVM classifier, reaching an accuracy of 98.4% and an AUC of 99%. However, our focus is on convolutional neural networks, which are particularly adept at analyzing spatial and temporal patterns in fMRI data. Researchers have explored various convolutional neural networks (CNN) or multilayer perception (MLP) on image to ad prediction, or both [18, 19]: in [20], Patil et al. reviewed various models used for AD detections using CNNs, where they found out that a 18 layer CNN model by Odusami [21] demonstrates an impressive accuracy of 98%; In [22], Chelladurai et al. proposed a MLP based network, where after their optimization method, it achieved an outstanding accuracy of 99.44%. These studies underscore CNN's robustness and adaptability in medical imaging, which are critical for developing clinically applicable AD diagnostic tools. In [23], Generative Adversarial Network (GAN) is applied to learn from magnetic resonance imaging (MRI) scans at multiple magnetic field strengths to boost the performance on AD prediction. Last but not least, the study by Gupta et al. [24] demonstrates the effective adaptation of deep learning models, originally trained on natural 2D images, to analyze 3D MRI scans and enhancing the model's performance in brain age prediction and

Alzheimer's disease detection tasks. These represent promising development in the field of medical imaging for AD. Such advancements reinforce the promise of deep learning, particularly CNNs, in transforming AD diagnosis.

Transitioning from imaging to genetics, the field faces more complex challenges [25]. For instance, Maj et al. [26] demonstrate an innovative approach by integrating deep learning with traditional machine learning methods to analyze gene expression data related to AD. On the other hand, [27, 28] both utilized gene expression data on AD risk prediction, while [28] also utilized DNA methylation data, both of the work indicated the difficulty in predicting AD risk using gene-related information. Authors of [28] also performed novel application of generative adversarial networks for RNA-seq analysis for molecular progress. In [29], a MLP framework is applied for predicting AD-specific sites using RNA data. These studies collectively highlight the dynamic nature of AD research, where advancements in one area, such as imaging, pave the way for new challenges and opportunities in other areas, like genetic analysis. This highlights the significant challenges of genetic predictions and reinforces the importance of imaging techniques, where CNNs offer a more direct and interpretable pathway for analyzing AD-related changes in brain structure.

Given the successes and challenges, our study initially aimed to develop predictive models using both genetic and fMRI data for a combined prediction. However, we found that incorporating genetic data did not enhance the model's performance significantly—it yielded unstable results as those obtained with fMRI data alone (as provided in the appendix). Consequently, our focus shifted to exclusively utilizing fMRI data. In this work, we present a Convolutional Neural Network (CNN) model to analyze fMRI scans and categorize participants as normal control (NC) or with AD. This approach is consistent with the current trajectory in AD research, where deep learning models are increasingly applied to neuroimaging data, as evidenced in studies like [30–32]. The primary knowledge gap our study addresses is the novel application of 3D CNNs to fMRI data for AD prediction. Unlike previous studies that primarily focused on detection accuracy, our research extends these efforts by introducing interpretable aspects. This not only enhances the understanding of model decisions but also addresses the often underexplored aspect of explainability.

While it is notable that not all recent research in this topic has publicly available code, we commit to ensure our code is easily accessible to other researchers. By comparing our findings with a select group of contemporary studies and making our code publicly accessible on GitHub, we aim to contribute to the collective effort in AD research.

## Methods and datasets

### Data availability statement

The data for this study are sourced from the Alzheimer's Disease Neuroimaging Initiative (ADNI), a longitudinal multicenter study aimed at identifying clinical, imaging, genetic, and biochemical biomarkers for AD detection and tracking [33]. The ADNI encompasses several phases: ADNI 1 (2004—2010), ADNI Go (2009—2011), ADNI 2 (2011—2017), and ADNI 3 (2017—2022). Our study primarily utilized data from ADNI 1, ADNI Go, and ADNI 2. The ADNI 2/Go phase includes participants from ADNI 1, some of whom may have experienced progression in their AD status, as shown in Fig 1. These data include de-identified fMRI scans and associated demographic and clinical assessments.

1. **Description of the data set and the third-party source**: The dataset includes fMRI scans along with demographic information such as age and gender, and clinical data related to Alzheimer's Disease status (normal control, mild cognitive impairment, or Alzheimer's

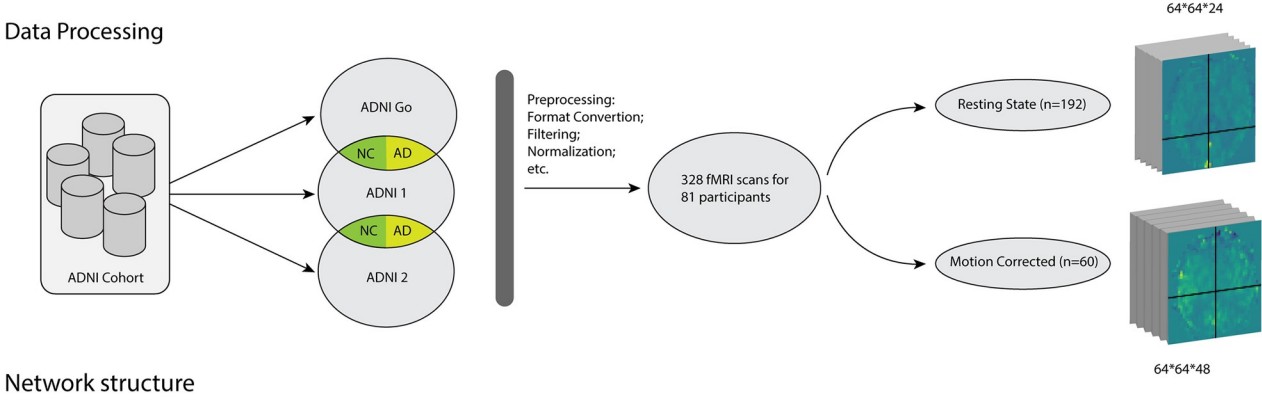

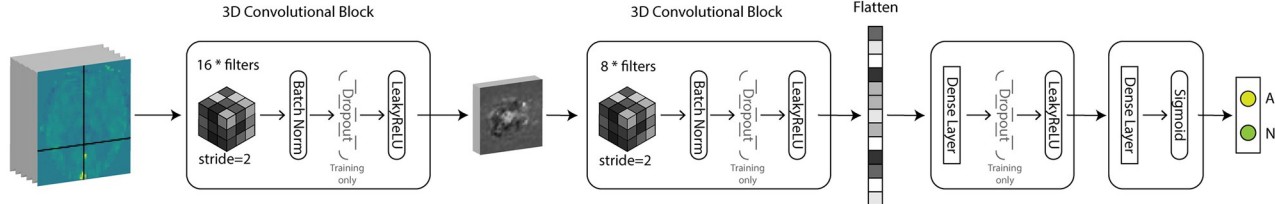

**Fig 1. Comprehensive diagram of neuroimaging study.** Upper section details data processing techniques including format conversion and normalization, while the lower section depicts the neural network structure with layers like 3D Convolutional Blocks and LeakyReLU. The diagram also includes information of the ADNI Cohort we used, with specific scan dimensions.

Disease). All data were obtained from the Alzheimer's Disease Neuroimaging Initiative (ADNI).

2. **Verification of permission to use the data set**: The use of ADNI data is governed by the ADNI Data Use Agreement, which all researchers must accept before accessing the data. Our study complied with all terms of this agreement.

3. **Confirmation of whether the authors received any special privileges in accessing the data that other researchers would not have**: No special privileges were granted to the authors in accessing the ADNI data. All data used in this study are available to other researchers under the same terms through the ADNI database.

4. **All necessary contact information others would need to apply to gain access to the data**: Researchers interested in accessing ADNI data should visit the ADNI website at https://adni.loni.usc.edu/data-samples/adni-data/#AccessData. The website provides detailed instructions on the registration and data access application process.

## Ethics statement

This study involves secondary analysis of existing public data obtained from the Alzheimer's Disease Neuroimaging Initiative (ADNI). The original collection of the data was approved by the institutional review boards of all participating institutions. Informed consent was obtained from all participants involved in the study. This secondary analysis did not involve direct interaction with human participants, and therefore no additional ethical approval was required for this study.

## Participants

Initially, we planned to create three corresponding datasets for different model types: fMRI as for CNN, gene for MLP, and merged for the combined model. However, due to challenges encountered with the gene data, our focused turned on the fMRI dataset. This dataset is comprised of 328 fMRI scans from 81 participants, where all are obtained from ADNI 2/Go for better input consistency. The fMRI data were categorized based on the number of slices into two types: Resting State fMRI (Resting State n = 192) and perfusion weighted fMRI (PW n = 60). Both categories of the dataset are divided into training (60%), validation (20%), and testing (20%) subsets for training and evaluating the model.

Table 1 presents the demographic information of the participants in the fMRI cohort we used. All analyses were conducted on de-identified data. The fMRI scans were downloaded from the ADNI portal (https://adni.loni.usc.edu/), and the current diagnosis (DXCURREN or DXCHANGE column) from the most recent visit was used for each participant [33].

## Data preprocessing

**fMRI preprocessing.** For the fMRI dataset, preprocessing is performed for standardizing the data and format for the following deep learning application and analysis. As the first step, the scans are converted from their original DICOM format to 4D NIfTI images using the dicom2nifti package, which then will be filtered based on the number of slices available: only fMRIs with 6720 slices in resting state and fMRIs with 405 slices in Motion Corrected Time Series are retained for future use. Scans that does not meet the criteria are excluded for consistency purpose, which enables more accurate and direct comparisons and analysis across different types of scans. The details and code for preprocessing are available in our GitHub repository (https://github.com/gersteinlab/fmgene/tree/master).

**Gene preprocessing.** Preprocessing steps were also undertaken for the gene dataset, which involved selecting a subset of participants, merging their genetic data, and performing feature selection to identify SNPs most correlated with AD status. Specifically, we first randomly selecting 100 participants out of 1093 and merging their genetic data from CSV files, where NA columns were ignored. The AD status of the participants is determined by their most recent visit, regardless of previous visit (if any). Consequently, a chi-square test was performed between the minor allele numbers of each SNP and the AD status to identify the 500 SNPs most correlated with AD. It worth noting that, not all of the 1093 patients have both gene data and imaging data available. While this process was initially undertaken, the findings from the gene data analysis are not included in the final study results due to the challenges encountered with the gene data, including not enhancing the model's predictive performance when combined with the fMRI data, and the highly imbalanced dataset. However, we still

Table 1. Demographic information of ADNI fMRI cohort.

| ADNI Participant Demographic Information for fMRI Cohort | |
|---|---|
| **Subject Characteristic** | **fMRI (n = 81)** |
| Age (mean) | 74.8 |
| Female (n, (%)) | 40 (49.38%) |
| Male (n, (%)) | 41 (50.62%) |
| CN (n, (%)) | 31 (38.27%) |
| MCI (n, (%)) | 44 (54.43%) |
| AD (n, (%)) | 6 (7.41%) |

include the codes that are related to the gene modeling in our code base (see the following section), and the experiment results related to gene data modeling in the appendix.

## Deep learning frameworks

**Availability.** The models for this study were developed primarily using Python and the PyTorch [34] package. The source code and implementation details can be easily accessed on the GitHub repository: https://github.com/gersteinlab/fmgene/tree/master.

**Baseline models.** To provide a robust comparison, we utilized several baseline models, each designed to address different aspects of the data characteristics:

- **RNN Model:** The baseline Recurrent Neural Network (RNN) model is tailored to capture temporal dependencies within the fMRI data sequences. It utilizes gated recurrent units (GRUs) to address the vanishing gradient problem, making it suitable for learning from time-series data.

- **MLP Model:** The baseline Multilayer Perceptron (MLP) using a standard fully connected network architecture to process static features extracted from fMRI data.

**Trainable parameters.** Here, we provide details on the trainable parameters for each model, demonstrating the computational efficiency of our proposed approach:

- **our Model:** Approximately 1.3 million (1,331,402) parameters.

- **Gupta's Model:** Approximately 2.9 million (2,948,866) parameters.

- **RNN Model:** Approximately 12.6 million (12,595,586) parameters.

- **MLP Model:** Approximately 25.2 million (25,174,338) parameters.

**Unified CNN architecture for fMRI data.** CNNs are known for their efficacy in image analysis [35], which employ convolutions to extract significant features relevant to the target output. This kind of network can identify the features of input by using multiple layers and filters, while also avoiding the overfitting issue via using filters, which can significantly reduce the number of neurons required than other kinds of networks. Therefore, it is especially popular when study on imaging data like in this work's setting. Here, we utilized a standardized Convolutional Neural Network (CNN), which is tailored so it can process different datasets from ADNI (Resting and Perfusion Weighted) uniformly. These data are divided into two classes: AD and NC, as mentioned before. With the fMRI as input, the model will outputs probability for each of these 2 classes. The hyperparameters and structure of the network are optimized based on empirical experience combined with current experimental results, and was evaluated via multiple experiments with different seeds. Details on the specific architecture and parameters of our CNN model are provided in the Supplementary Information. Our unified CNN architecture is designed to efficiently process varying fMRI data while maintaining consistency in feature extraction and classification methodologies. A graphical reference for visualization of the network can be referred to at Fig 1.

**Comparison of model architectures.** In our study, we have developed a model, hereafter referred to as "ours," and conducted a comparative analysis with the model developed by Gupta et al. [24], hereafter referred to as "Gupta's." Their work provides a valuable point of comparison for our study.

Gupta's model employs a series of 3D convolutional layers, each accompanied by instance normalization and pooling layers, designed to incrementally increase the number of filters

for extracting increasingly complex features. This design is concluded with an average pooling layer and a final convolutional layer, leading to a sigmoid function for binary classification.

In contrast, our model exhibits several different architectural choices. We have a different number of layers (2 compared to Gupta's 5), which influences the depth and complexity of feature extraction. Our approach to normalization utilizes batch normalization, which aims to enhance model generalization by normalizing features across all samples. Notably, instead of conventional pooling layers, our model uses strides in the convolutional layers. This choice allows the model to learn the most effective feature extraction method dynamically, rather than relying on a fixed pooling function. This can be particularly advantageous in preserving the spatial resolution of fMRI data features, which could be important in medical imaging where detailed information is needed. Furthermore, the integration of dropout layers in our model acts as a regularization strategy to prevent overfitting, thereby improving the model's ability to generalize to unseen data.

**Evaluation metrics and tools.** To ensure a comprehensive assessment of our model's performance, we employed several evaluation metrics and tools, which are detailed below:

**Receiver Operating Characteristic (ROC) curves.** The Receiver Operating Characteristic (ROC) curve is a graphical plot that illustrates the diagnostic ability of a binary classifier system as its discrimination threshold is varied. It is used to plot the True Positive Rate (TPR, sensitivity) against the False Positive Rate (FPR, 1-specificity) for different threshold settings. The Area Under the ROC Curve (AUC) provides a single measure of overall model performance across all classification thresholds, with a value of 1 representing perfect accuracy and 0.5 denoting a result no better than chance.

**Performance metrics.** The following metrics were calculated to evaluate the performance of our deep learning models:

- **Accuracy**: The proportion of true results (both true positives and true negatives) among the total number of cases examined.

- **Precision**: The ratio of correctly predicted positive observations to the total predicted positives. It is a measure of the accuracy provided that a specific class has been predicted.

- **Recall (Sensitivity)**: The ratio of correctly predicted positive observations to all observations in actual class.

- **F1 Score**: The weighted average of Precision and Recall. This score takes both false positives and false negatives into account.

- **Matthews Correlation Coefficient (MCC)**: A coefficient that produces a high score only if the prediction obtained good results in all of the four confusion matrix categories (true positives, false negatives, true negatives, and false positives), relative to the size of positive elements and negative elements in the dataset.

**Interpretability tools.** *SHAP (SHapley Additive exPlanations)*: SHAP values interpret the impact of having a certain value for a given feature in comparison to the prediction. This approach originated from game theory and provides a foundation for interpreting model output. In our study, SHAP values were utilized to highlight the most influential features in our model's predictions, potentially identifying key brain regions in fMRI scans related to Alzheimer's Disease.

## Results

### Model performance

We performed 5 experiments with different seeds to measure the model's performance, which in turn results in different initial weights (using Kaiming Initialization) and different splits of training, testing and validation sets. We also introduced the deep learning model developed by [24] as a comparison with contemporary SOTA framework on AD prediction. To ensure the most objective comparison, we employed the same split of dataset and initialization seed for both of the models, with same training epochs, and other hyperparameters of the network as in their original paper. It worth noting that, while there are recent papers (published after 2021) using deep learning (often CNNs) with fMRI/MRI as input for predicting AD risk that provides valuable insights into the field, the availability of their code is not commonly disclosed or readily available in the public domain due to different reasons (i.e. intellectual properties, privacy reasons, etc.). In our research, due to the focus on accessible resources, we have primarily drawn comparisons with the model proposed by Gupta et al. in [24]. This approach not only aligns with our research objectives but also allows us to build upon and contribute to the body of work with openly available resources. We believe that our study complements these recent advancements and collectively. Further, we aim to introduce more comparisons, should we found more recent works that are related to this study, and with publicly available source codes for their frameworks.

In Table 2, we present the comparative results of various deep learning models' performance on the Resting dataset and the PW dataset, which includes our CNN, Gupta's CNN [24], and two baseline models (RNN & MLP). The table contains accuracy, precision, recall, F1, Matthews correlation coefficient (MCC), and area under the curve for a comprehensive evaluation.

In the Resting dataset, our model exhibited competitive performance, achieving an Accuracy and AUC of **0.8333 (±0.0589)** and **0.8930 (±0.0588)** respectively. These results were slight better than those of Gupta's model [24], as our model demonstrated superior Precision, Recall, F1, and MCC. The baseline models, particularly the Baseline_MLP, showed significantly lower performance across all metrics, underlining the complexity of the dataset and the effectiveness of our proposed approach.

**Table 2. Comparison of classifier performance.**

| (a) Performance on Resting Dataset | | | | |
|---|---|---|---|---|
| Model | Accuracy | F1 | MCC | AUC |
| our | **0.83 (±0.06)** | **0.17 (±0.29)** | **0.17 (±0.29)** | **0.89 (±0.06)** |
| Gupta's [24] | 0.83 (±0.06) | 0.00 (±0.00) | 0.00 (±0.00) | 0.88 (±0.07) |
| Baseline_RNN | 0.81 (±0.07) | 0.00 (±0.00) | -0.03 (±0.06) | 0.42 (±0.11) |
| Baseline_MLP | 0.44 (±0.44) | 0.00 (±0.00) | 0.00 (±0.00) | 0.50 (±0.00) |
| (b) Performance on Perfusion Weighted (PW) Dataset | | | | |
| our | **0.93 ±0.04)** | **0.19 ±0.26)** | **0.22 (±0.28)** | **0.95 (±0.03)** |
| Gupta's [24] | 0.92 (±0.04) | 0.00 (±0.00) | 0.00 (±0.00) | 0.76 (±0.26) |
| Baseline_RNN | 0.92 (±0.04) | 0.00 (±0.00) | 0.00 (±0.00) | 0.54 (±0.06) |
| Baseline_MLP | 0.36 (±0.42) | 0.02 (±0.04) | 0.00 (±0.00) | 0.53 (±0.04) |

This table presents a comparative analysis of different machine learning models on two datasets: Resting and PW. The performance metrics include Accuracy, Precision, Recall, F1, Matthews Correlation Coefficient (MCC), and Area Under the Curve (AUC). The results highlight the varying effectiveness of each model, especially in the context of a highly imbalanced dataset. Note that the best results in each metric are bolded for clarity. This comparison aims to shed light on the models' capabilities in handling imbalanced data and the relative importance of each metric in such scenarios.

The performance on the PW dataset suggests the strength of our model. Here, our model clearly outperformed all others in every metric, achieving an Accuracy of **0.9282 (±0.0410)** and an AUC of **0.9502 (±0.0258)**. Gupta's model and Baseline_RNN were close in terms of Accuracy, but our model demonstrated a higher degree of precision in predictions and other metrics. The Baseline_MLP, again, lagged behind, particularly in terms of Accuracy and AUC.

The precision, recall, F1 are low, particularly because of the small amount of data available, and the highly unbalanced labels (i.e. approximately 9: 1 ratio). Firstly, the small dataset size inherently limits the model's ability to learn and generalize effectively. Less examples introduce additional difficulty for the model to capture the underlying patterns of the data, leading to a decreased ability. Secondly, the label imbalance, with a ratio of approximately 9:1, further exacerbates this issue. The overwhelming majority of one class in the dataset can lead to a model bias towards predicting the majority class. This imbalance skews the model's learning process, often resulting in poor performance in identifying the minor class. In consequence, metrics such as recall (which measures the model's ability to identify all relevant instances), precision (which assesses the accuracy of these identifications) are significantly affected. Similarly for F1 score, as it is basically the weighted mean of the precision and recall. Overall, these metrics highlight the need for a more balanced dataset and a larger sample size to improve the model's performance and reliability, especially in complex and sensitive applications. On the other hand, the results demonstrates our model's advantage in handling small and highly imbalanced datasets, a relatively common challenge in medical data analysis. The consistency of performance across both datasets suggests that our approach is comparatively robust and adaptable. It is also noteworthy that our model maintained relatively high performance in terms of both AUC and Accuracy across the two datasets.

We also performed statistical analyses of our and Gupta's model performances on different datasets, as summarized in Table 3. For the Resting test dataset, the T-test (T-stat = -1.96, p = 0.0545) suggested a trend towards a difference in model performance that did not reach conventional statistical significance. However, the significant result from the Wilcoxon test (W-stat = 311, p = $8.73 \times 10^{-6}$) indicates a consistent difference in the median of the ranked differences, suggesting that one model generally outperformed the other. For the PW dataset, both the T-test (T-stat = -3.07, p = 0.0025) and the Wilcoxon test (W-stat = 6649, p = $2.31 \times 10^{-4}$) showed significant differences in model performance, indicating consistent discrepancies between the models across this dataset. The significant values from both the T-test and Wilcoxon test for the PW dataset confirm robustness in performance, suggesting that our model is more suited to external conditions represented by this dataset, while performs as well as the Gupta's model.

## ROC curve

The visual reports for the performance comparison of the fMRI models are presented in Fig 2. In this figure, ROC curves for all four models are plotted for a direct comparison. For each model, the Area Under the Curve (AUC) was computed using the mean of the results from the

**Table 3. Statistical comparison of model performance on resting and PW datasets.**

| Dataset | T-test (T-stat, P-value) | Wilcoxon Test (W-stat, P-value) |
|---|---|---|
| Resting Test | -1.96, 0.0545 | 311, $8.73 \times 10^{-6}$ |
| PW Dataset | -3.07, 0.0025 | 6649, $2.31 \times 10^{-4}$ |

The table presents T-statistics and p-values for the T-tests alongside W-statistics and p-values for the Wilcoxon tests, indicating differences in model performance across datasets.

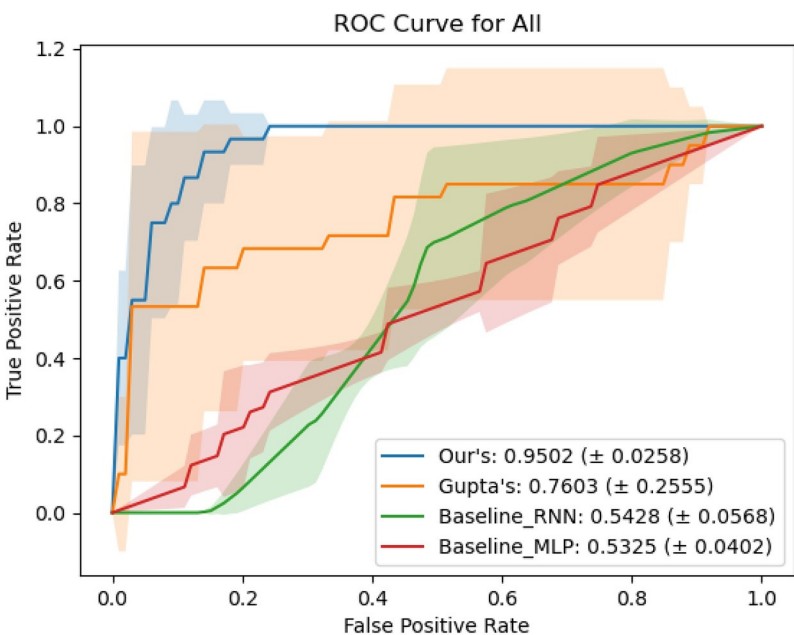

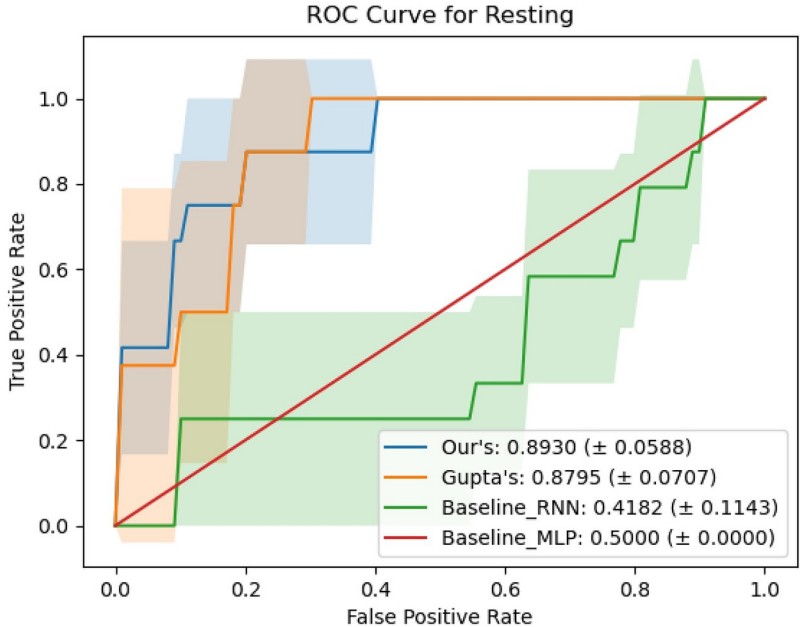

**Fig 2. Comparative ROC curves for neural network models.** Receiver Operating Characteristic (ROC) curves comparing the performance of different neural network models, including Our CNN, Gupta's CNN, Baseline RNN, and Baseline MLP, across different datasets. Each curve represents the trade-off between the True Positive Rate (TPR) and False Positive Rate (FPR) for a specific model, providing insights into their diagnostic ability in distinguishing between classes.

five experiments. The corresponding shaded areas represent the standard deviation at each stage, illustrating the variability across different experiments.

Based on these figures, it is evident that the performance of our proposed model generally surpasses that of the other models. However, it's noteworthy that in some individual seeded

experiments, Gupta's model [24] may exhibit superior performance. This suggests a certain level of variability in model performance depending on the specific conditions of each experiment.

In summary, our experiments demonstrate the superiority of our model in handling complex and imbalanced datasets typical of AD predictions. While comparisons were limited due to the availability of source codes and time constraints, our findings offer insights into the potential of advanced deep learning techniques in medical diagnostics. Future work will aim to include more comprehensive comparisons with other contemporary models, enhancing the robustness and applicability of the framework.

### Interpretability

Understanding which features are important in predicting Alzheimer's Disease (AD) is as important as the classification accuracy analysis of AD risks and imaging data. For this purpose, we have made visualizations of feature maps for analysis of the feature extracted by the CNN model, and employed SHAP (SHapley Additive exPlanations) values visualization to identify the most influential voxels in the brain scans. For further comparative analysis between our proposed model and Gupta's model, we have included additional plots in the supplementary materials, please refer to supplementary document for details.

The feature map visualizations presented in Fig 3 are applied to the first layer of the convolutional neural network. These visualizations represent max-pooled and normalized feature maps derived from the brain scans of patients. Each map corresponds to a unique filter applied within the layer, which capturing different aspects of the input data. For patients with and without Alzheimer's Disease, these maps highlight distinctive patterns and intensities, suggesting areas that the network deems significant for the classification task. The variance in activation across these feature maps illustrates how the convolutional network processes and

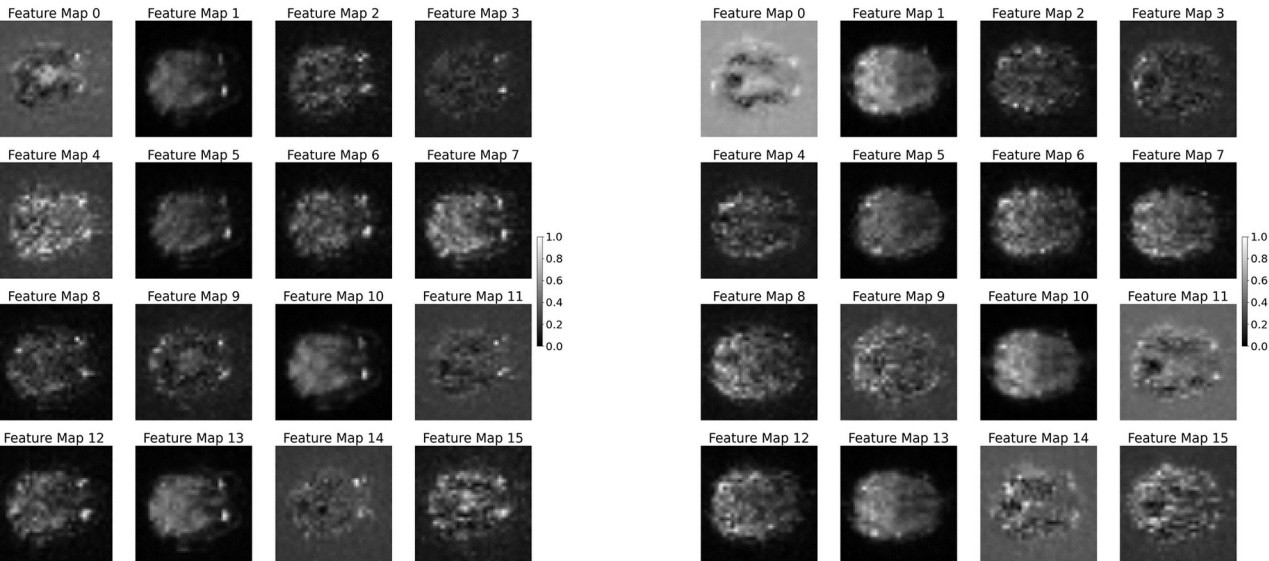

**Fig 3. Comparison of max-pooled and normalized feature maps from the first convolutional layer of our proposed model for AD prediction.** On the left, feature maps from a participant diagnosed with Alzheimer's Disease (AD) show distinct patterns of activation in regions associated with the condition. On the right, feature maps from a cognitively normal (NC) participant exhibit different attention to activations, highlighting differences leveraged by the CNN for diagnostic classification.

differentiates between the structural nuances of AD-affected brains compared to those of healthy controls. These visualizations provide intuitive understanding of the model's focus areas and decision-making process.

For example, in Fig 3, there are various noticeable differences in the intensity of activations between patients with and without Alzheimer's Diseas that may be of interesting to inspect. The higher activation values in could reflect the model's capability to recognize the important signs of AD. The pronounced activation in these areas may reflect the CNN's capability to recognize the key signs of AD, aligning with the known neuropathological hallmarks of the disease. Given that these are feature maps from the first convolutional layer, they likely represent initial patterns and contrasts identified by the model from the raw fMRI data. The presence of higher activation in these feature maps could indicate the network's detection of atypical patterns associated with structural changes or abnormalities in brain regions affected by AD.

In addition, we overlay SHAP values onto the grayscale fMRI images, as shown in Fig 4 to highlight the voxels that are considered important by the network [36]. This visualization method highlights the most influential voxels in the brain scans according to their contribution to the model's output. The SHAP values, represented as color-coded pixels, range from 0 to 1, with warmer colors indicate regions that have a significant influence on the model's output. This influence pertains to identifying both the presence of AD characteristics and the absence thereof, as shown in the contrasting patterns observed in NC and AD patients. For a better rendering, we only selected the top 3% pixels based on their impact weights to the prediction results.

Upon examining the SHAP value distribution in Fig 4, we identify key regions where the model discerns differences between NC and AD patients. For instance, clusters of high SHAP values are evident in the temporal and parietal lobes [37], regions implicated in AD pathophysiology. Specifically, our observations indicate higher activations on hippocampus in NC patients, compared to those with AD. These prominent activations in NC cases may signal to the model features of a healthy hippocampus, thereby influencing its classification towards a NC diagnosis. This pattern aligns with the understanding that a non-diseased hippocampus typically does not exhibit the structural and functional changes associated with AD, suggesting that the model is effectively identifying key characteristics that differentiate between the two states. The hippocampus is known to be involved in memory [38] and attentional processes [39], and its dysfunction has been well discussed in related researches [38, 39] (a reference image where hippocampus area is highlighted are available in Fig 5, note this figure is a

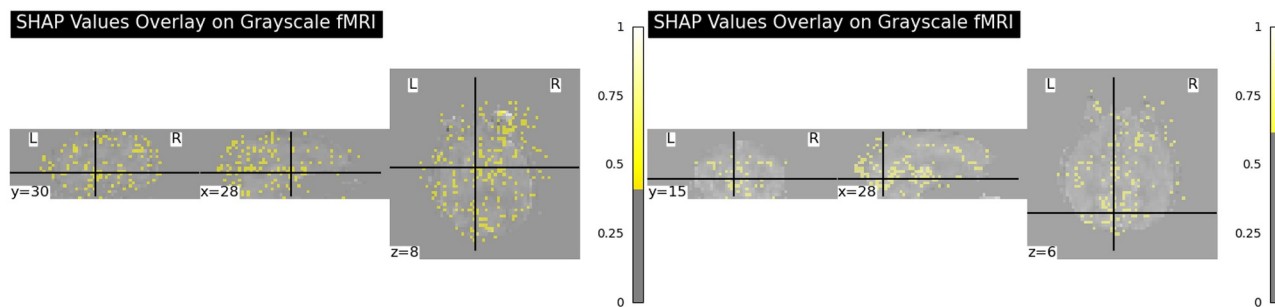

**Fig 4. Visualization of SHAP values on grayscale fMRI scans.** The left image depicts the spatial distribution of influential voxels for a patient with Alzheimer's Disease, and the right image for a NC subject. The overlays highlight the voxels that most significantly contribute to the CNN model's predictive differentiation between AD presence and absence. These visualizations indicate potential regions impacted by AD and the model's classification strength.

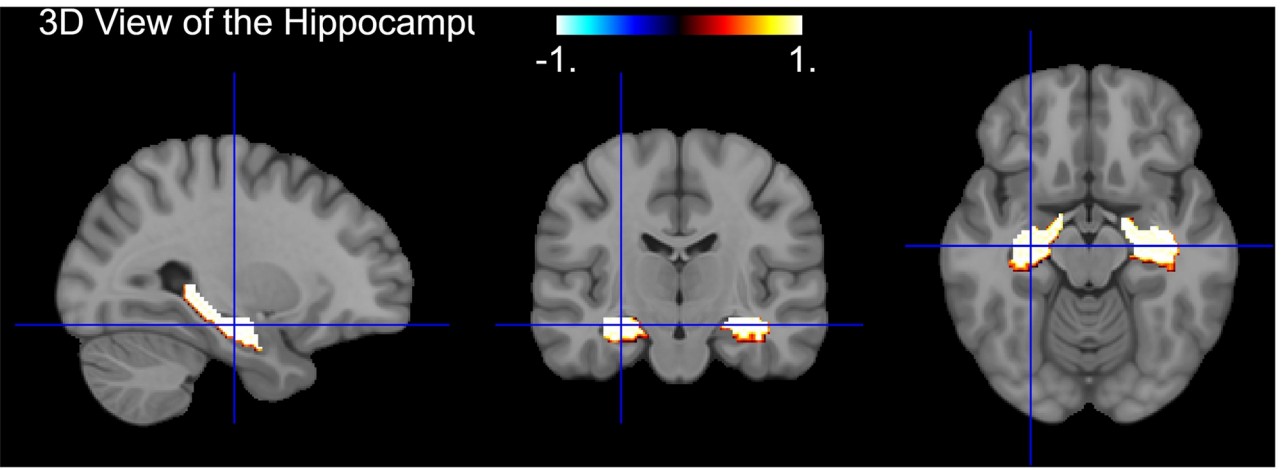

**Fig 5. Multi-angled fMRI scans highlighting regions of hyperactivation in the hippocampus of an Alzheimer's Disease patient.** The color overlay indicates areas of hippocampus region, which are known to related to AD. The left panel shows a sagittal view, the center panel depicts a coronal view, and the right panel illustrates an axial view, together providing a comprehensive 3D perspective of hippocampal engagement.

template). These areas are critical for cognitive functions such as memory and spatial orientation, which are often impaired in AD patients. On the other hand, the brightness and highlighted points are noticeable less in the normal control case, compared to the AD case. The visualization shows that the model is focusing on related regions, lending credibility to its predictions and providing a visual explanation for the model's decision-making process. By overlay SHAP values onto the brain scans, we can not only validate the model's accuracy but also gain valuable insights into the anatomical information of AD. This could potentially guide the medical community in early diagnosis and intervention strategies.

Finally, we present the SHAP Maximum Intensity Projection in Fig 6, which aggregated the most significant pixels across different slices. Thus, the resulting heatmap displays the regions that are considered the most substantial for AD, serve as a supplementary interpreting perspective for the AD association with the fMRI.

## Discussion

The findings of this study underscore the potential of deep learning, particularly Convolutional Neural Networks (CNNs), in predicting Alzheimer's Disease using fMRI data. Our CNN models, while surpassing the listed previous SOTA deep learning model and other baseline models for AD prediction, also keeps a concise structure, which requires less training time due to the less amounts of neurons needed. This suggests that even with relatively simpler architectures, deep learning can be quite effective in AD prediction when settled up appropriately. In addition, it indicates that complex models may not always be necessary for accurate AD prediction, especially when established in the common constraints of medical datasets, such as size limitations and label imbalance.

The study advances the explainability of predictive models in AD research. Visualization of feature maps from our CNN models has shown activation patterns that align with established areas of hyperactivation in AD. For instance, it indicates that medial temporal lobe, which includes core structures that are related to memories, etc., are related to AD risk more than other areas. This alignment with prior medical knowledge indicates that our model correctly prioritizes the features in the brain scans for AD prediction. Besides, our presentation of

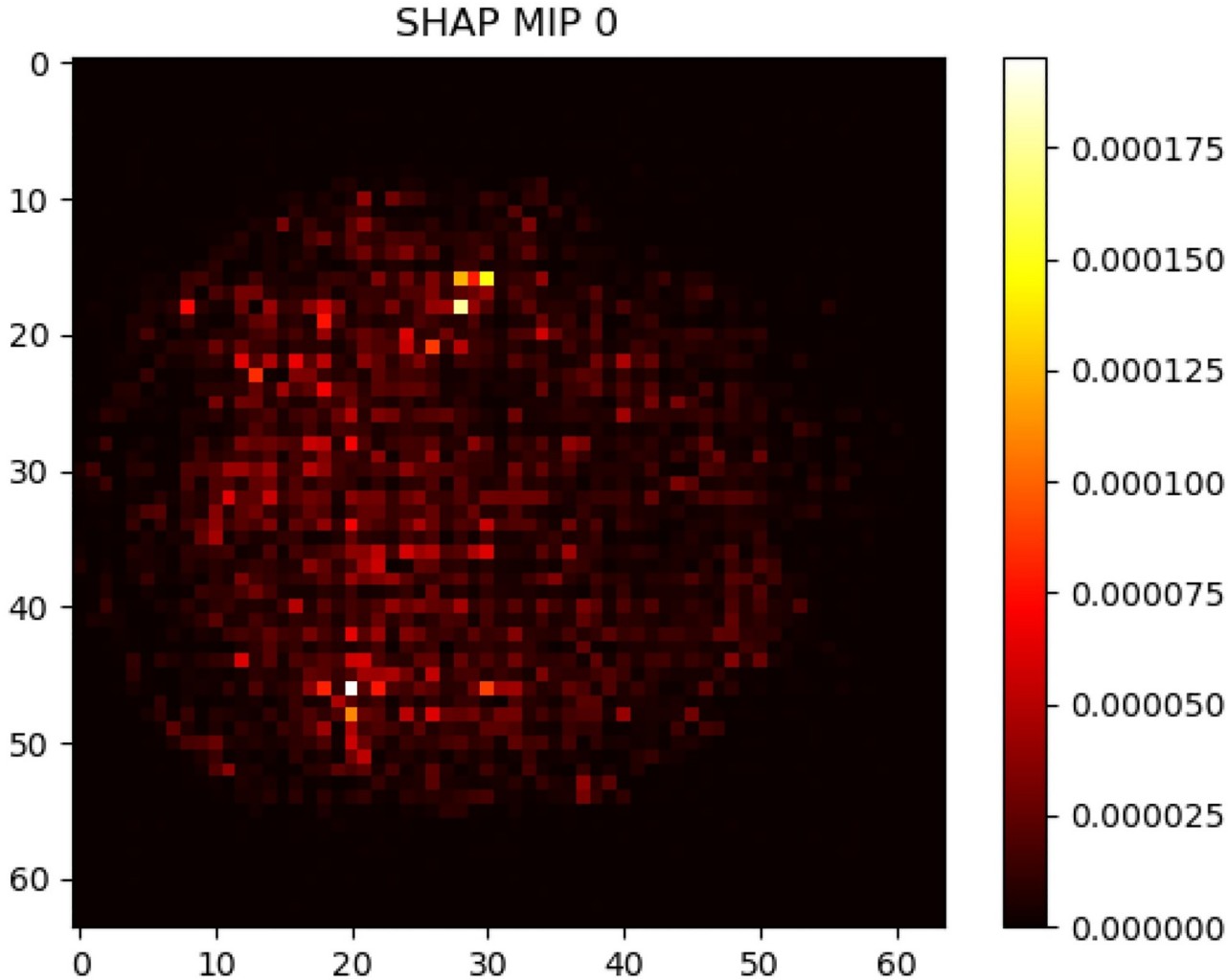

**Fig 6. SHAP Maximum Intensity Projection (MIP) for Alzheimer's Disease prediction.** The heatmap illustrates the concentration of SHAP values across various brain slices, with brighter colors indicating higher importance in the model's predictive assessment. This MIP view consolidates the most significant voxels, offering a comprehensive perspective on the regions critical for AD classification.

SHAP overlays on the fMRI helps gaining insights to understanding the individual contributions of voxels to the AD risk evaluation. As a portion of the highlighted regions aligns with established medical knowledge in AD pathology, the findings for CNN visualization may also provide insights into previously unidentified brain areas that could be associated with AD. This could pave the way for novel hypotheses and further investigations of the disease. Furthermore, the Maximum Intensity Projection visualizations synthesize SHAP insights into a single two-dimensional representation, offering a more general overview of the influential brain regions for a more general analysis and understanding.

## Areas for improvement

The primary focus on distinguishing between CN and AD subjects in our study was driven by the objective to enhance early detection mechanisms for AD using neuroimaging data. While we recognize that incorporating Mild Cognitive Impairment (MCI) as a distinct class could

offer more granular insights into the progression stages of AD, our current dataset and model aimed to first establish a robust framework for the more binary distinction relevant in initial screenings. The limitations posed by the dataset size and balance issues further constrained the feasibility of including MCI without risking the model's ability to learn effectively. However, expanding the classification to include MCI could substantially increase the clinical applicability of our model. The potential to improve clinical applicability can be expanded to **include MCI** using a larger dataset for more granular insights and a tri-class classification system. This advancement would not only improve the model's sensitivity to AD-specific neuroimaging biomarkers but would also incorporate domain-specific knowledge to significantly enhance the predictive accuracy and clinical applicability of the findings.

On the other hand, the interpretability analysis may be perceived as speculative without direct clinical validation. Recognizing the importance of involving clinical experts in neurodegenerative diseases, the integration of 'expert-in-the-loop' systems could be beneficial for future research as resources allow. This enables identifying and mitigating biases and ensuring that the model's focus aligns with clinically relevant features, thereby potentially enhancing the predictive accuracy and clinical applicability of the findings.

While our current approach has shown promise, one area for future improvement involves the exploration of data augmentation techniques. In this study, we opted not to use such methods due to potential concerns about introducing non-authentic variations into the complex fMRI data. However, developing and testing sophisticated augmentation strategies that preserve the integrity of the underlying biological signals could potentially improve model performance, especially in addressing class imbalance. This area warrants further investigation to find a balance between maintaining data authenticity and enhancing the dataset to train more robust models.

Additionally, **expanding the fMRI dataset** is crucial for increasing the robustness and generalizability of our results. Incorporating longitudinal data would provide valuable insights into the progression of AD, thereby augmenting the predictive capabilities of our models over time. As all the data currently utilized are sourced from ADNI, integrating **external datasets** for validation could substantially bolster the model's reliability and its applicability across diverse demographic and clinical settings.

Moreover, addressing class imbalance remains a critical challenge in the effective training of predictive models [40]. In this study, we opted not to use data augmentation techniques to mitigate class imbalance, due to concerns about the authenticity and representation of the complex fMRI data. This decision, while aligned with our goal to preserve data integrity, suggests an area for improvement. Future research could explore the development and validation of sophisticated data augmentation techniques that are capable of enhancing dataset balance without compromising the quality and reliability of the data. This exploration is crucial for improving model performance and ensuring its clinical viability.

The **integration of multi-modal data**—merging fMRI with genetic and clinical data—presents a promising strategy to enrich the study's insights. For instance, genetic data could unveil predispositions to Alzheimer's Disease, while clinical assessments might contextualize the neuroimaging findings, leading to a more nuanced understanding of the disease's dynamics and impact.

Lastly, the exploration of **advanced or novel algorithms** holds the potential to enhance predictive performance further. Investigating methods to address imbalanced datasets, such as the application of synthetic data generation techniques like generative adversarial networks (GANs) or employing advanced sampling methods, could significantly refine model efficacy. Moreover, in furthering the efficacy of AD detection using fMRI scans, an architecture specifically optimized for AD-related neuroimaging data could refine the model's ability to detect nuanced patterns specific to AD.

## Conclusion

This study contributes to the intersection of deep learning and Alzheimer's Disease diagnosis. It demonstrates that a compact neural network design can not only effectively analyze fMRI data for AD prediction, but also supports better interpretability via reduced number of layers and complexities. Moreover, the correlation between the focused areas of our model and the brain regions known to be affected by AD reinforces the viability of employing convolutional neural networks for medical imaging analysis in neurodegenerative disease research. As the medical community advances towards the goals of early diagnosis and personalized treatment for AD, the utilization of interpretable deep learning models becomes increasingly important. These models hold promise for advancing our approach to detecting and understanding neurodegenerative diseases, potentially leading to more effective interventions, understanding of disease pathology, and improved patient outcomes.

## Supporting information

**Network parameters.** The specific architecture of our CNN model is structured as below:

**Convolutional Layers:** The model begins with two 3D convolutional layers designed to process the volumetric nature of fMRI data, capturing spatial relationships in three dimensions. The first convolutional layer applies a kernel of size 3x3x3 and with stride of 2x2x2 (to replace the pooling layer), which helps in reducing the dimensionality of the data while capturing detailed features. This layer is followed by 3D batch normalization, which stabilizes and accelerates the training process, and the LeakyReLU activation function. LeakyReLU introduces non-linearity and helps address the vanishing gradient problem, allowing the model to learn more complex patterns. The second convolutional layer has a similar setting, with a smaller number of filters for additional feature extraction, aiding in capturing more nuanced information from the fMRI data.

**Fully Connected Layers:** After the convolutional layers follows two fully connected layers. The first layer's input features vary based on the dataset, allowing the model to adapt to different data sizes and complexities. The second layer is designed to predict the AD status based on the encoded feature vector. The final outputs will be the output after a Sigmoid function, which is appropriate for binary classification tasks, as it interprets the likelihood of AD status.

**Dropout Layers & Parameters:** Dropout layers are applied throughout the network, except in the last layer, to prevent overfitting. A dropout rate of 0.5 means there's a 50% chance that each neuron's output will be set to 0 during training, ensuring the model does not rely too heavily on any single feature. The learning rate is set to be 0.001, balancing the speed of convergence with the risk of overshooting the minimum. The binary cross entropy (BCE) loss function is appropriate for our binary classification task, and the stochastic gradient descent (SGD) optimizer with a weight decay (L2 norm) of 0.1 helps in updating the model's weights effectively while also regularizing the model.

*Included below are additional SHAP values, feature maps on fMRI data, and gene results for interested readers.*

**Gene-fused Model Results:** We also present results of the model with gene information as additional inputs below, which includes S1 Table for various statistics and a figure for AUCs on the two datasets.

## Supporting information

**S1 Fig. Comparison of max-pooled and normalized feature maps from the first convolutional layer for Alzheimer's Disease (AD) prediction.** The left panel displays feature maps

from a participant diagnosed with AD. The right panel shows feature maps from a cognitively normal (NC) participant. S1 of S3 Figs.
(TIF)

**S2 Fig. Visualization of SHAP values on grayscale fMRI scans for Alzheimer's Disease (AD) prediction.** (1/3) Each pair of images shows the spatial distribution of influential voxels: the left image for a participant diagnosed with AD and the right image for a cognitively normal (NC) participant. Through these comparative visualizations, potential AD-impacted regions and the model's diagnostic accuracy are highlighted.
(TIF)

**S3 Fig. Comparison of max-pooled and normalized feature maps from the first convolutional layer for Alzheimer's Disease (AD) prediction.** The left panel displays feature maps from a participant diagnosed with AD. The right panel shows feature maps from a cognitively normal (NC) participant.
(TIF)

**S4 Fig. Visualization of SHAP values on grayscale fMRI scans for Alzheimer's Disease (AD) prediction.** Each pair of images shows the spatial distribution of influential voxels: the left image for a participant diagnosed with AD and the right image for a cognitively normal (NC) participant. Through these comparative visualizations, potential AD-impacted regions and the model's diagnostic accuracy are highlighted.
(TIF)

**S5 Fig. Comparison of max-pooled and normalized feature maps from the first convolutional layer for Alzheimer's Disease (AD) prediction.** The left panel displays feature maps from a participant diagnosed with AD. The right panel shows feature maps from a cognitively normal (NC) participant.
(TIF)

**S6 Fig. Visualization of SHAP values on grayscale fMRI scans for Alzheimer's Disease (AD) prediction.** Each pair of images shows the spatial distribution of influential voxels: the left image for a participant diagnosed with AD and the right image for a cognitively normal (NC) participant. Through these comparative visualizations, potential AD-impacted regions and the model's diagnostic accuracy are highlighted.
(TIF)

**S7 Fig. Feature maps from the first convolutional layer of Gupta's CNN.** The left panel displays feature maps from a participant diagnosed with AD. The right panel shows feature maps from a cognitively normal (NC) participant.
(TIF)

**S8 Fig. Visualization of SHAP values on grayscale fMRI scans by Gupta's model.** Each pair of images demonstrates the spatial distribution of influential voxels: the left image for a participant diagnosed with AD and the right image for a cognitively normal (NC) participant.
(TIF)

**S9 Fig. Comparative ROC curves for neural network models.** Receiver Operating Characteristic (ROC) curves comparing the performance of models with different input data, specifically our CNN with and without gene data as additional inputs across different datasets. Each curve represents the trade-off between the True Positive Rate (TPR) and False Positive Rate (FPR)

for a specific model.
(TIF)

**S1 Table. Comparison of classifier performance with merged datasets.** This table presents a comparative analysis focusing on the performance of our model with merged input data from Resting and PW datasets. The performance metrics include Accuracy, Precision, Recall, F1, Matthews Correlation Coefficient (MCC), and Area Under the Curve (AUC). The results showcase the models performance does not significantly improved with gene as additional input.
(TIF)

## Author Contributions

**Conceptualization:** Xiao Zhou, Mark Gerstein.

**Data curation:** Sanchita Kedia.

**Formal analysis:** Xiao Zhou, Sanchita Kedia.

**Investigation:** Xiao Zhou, Ran Meng.

**Methodology:** Xiao Zhou.

**Project administration:** Mark Gerstein.

**Resources:** Mark Gerstein.

**Software:** Xiao Zhou, Sanchita Kedia.

**Supervision:** Mark Gerstein.

**Validation:** Xiao Zhou, Sanchita Kedia, Ran Meng.

**Visualization:** Xiao Zhou.

**Writing – original draft:** Xiao Zhou, Sanchita Kedia, Mark Gerstein.

**Writing – review & editing:** Xiao Zhou, Ran Meng, Mark Gerstein.

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
