## [Decision Letter · Decision Letter 0]

26 Jul 2024

PONE-D-24-25018Deep learning analysis of fMRI data for predicting Alzheimer's Disease: a focus on convolutional neural networks and model interpretabilityPLOS ONE

Dear Dr. Gerstein,

Thank you for submitting your manuscript to PLOS ONE. After careful consideration, we feel that it has merit but does not fully meet PLOS ONE’s publication criteria as it currently stands. Therefore, we invite you to submit a revised version of the manuscript that addresses the points raised during the review process.

We look forward to receiving your revised manuscript.

Kind regards,

Xiaohui Zhang

Academic Editor

PLOS ONE

Journal Requirements:

2. For studies involving third-party data, we encourage authors to share any data specific to their analyses that they can legally distribute. PLOS recognizes, however, that authors may be using third-party data they do not have the rights to share. When third-party data cannot be publicly shared, authors must provide all information necessary for interested researchers to apply to gain access to the data. (https://journals.plos.org/plosone/s/data-availability#loc-acceptable-data-access-restrictions) 

4. We note you have included a table to which you do not refer in the text of your manuscript. Please ensure that you refer to Table 3 in your text; if accepted, production will need this reference to link the reader to the Table.

Reviewers' comments:

Reviewer's Responses to Questions

**Comments to the Author**

1. Is the manuscript technically sound, and do the data support the conclusions?

Reviewer #1: Yes

Reviewer #2: No

Reviewer #3: Yes

2. Has the statistical analysis been performed appropriately and rigorously? 

Reviewer #1: No

Reviewer #2: No

Reviewer #3: Yes

3. Have the authors made all data underlying the findings in their manuscript fully available?

Reviewer #1: Yes

Reviewer #2: Yes

Reviewer #3: Yes

4. Is the manuscript presented in an intelligible fashion and written in standard English?

Reviewer #1: Yes

Reviewer #2: Yes

Reviewer #3: Yes

5. Review Comments to the Author

Reviewer #1: This manuscript describes a new machine learning algorithm using convolutional networks to detect Alzheimer’s disease from fMRI scan data. The manuscript is well written and the following minor comments and suggestions should be addressed:

1. The authors should consider re-organizing their figure placement such that it appears closer to the section where they are discussed.

2. In the "Deep Learning Frameworks" section, more background information should be given on the baseline RNN and MLP models. The number of trainable parameters in all models should be listed for comparison.

3. Some statistical analysis should be added to see if the performance improvement is statistically significant compared to Gupta's model.

4. Did the authors use any data augmentation techniques to compensate the issue of class imbalance? Why or why not? The authors should provide some discussion on this.

5. Figure 3. The authors should indicate which model was used to generate this plot. If the authors' model was used, it might be worthwhile to compare that with a similar plot generated from Gupta's model for comparison.

Reviewer #2: - ## paper summary:

- This study aims to demonstrate feasibility of using fMRI + standard CNNs to predict risk of AD diagnosis with high performance despite low sample size and highly imbalanced training data. Post-hoc CNN feature map/SHAP explanations suggest that authors' CNN model relies on clinically known brain regions for assessing AD risk.

- ## strengths:

1) publicly available code

2) post-hoc model interpretation seems to align with domain knowledge of AD/MCI

- ## major issues:

1) the scientific goal, focus, or overall argument/hypothesis/structure of the study is not evident

- Q: why is the focus of the study to build a CNN? what purpose is it expected to serve given prior models with AUROCs/accuracies above 0.95/95%?

- the introduction is not thematically aligned with the running title of the paper

2) lack of overall scientific contributions:

- Q: what exactly is the knowledge gap in past literature that this study wants to address/explore or is motivated by?

- related literature and its limitations must be summarized under above context and not simply enumerated.

- Q: is the only issue that past models/code were not publicly released?

- Q: what novel insight is the reader supposed to take away from the interpretability analysis?

3) lack of clinical contribution:

- study maybe based on a flawed/irrelevant experimental design - not sure if CN vs AD is a clinically relevant classification task at all (although it is convenient for training a ML/DL classifier). Given that this model and others before it all have achieved very high AUC/accuracy, why are these models not yet trusted for use in real-world clinical workflows? Perhaps a design that focuses on CN vs MCI may be more clinically impactful?

- the interpretability analysis seems highly speculative/cherry-picked and affected by authors' confirmation bias, no expert-in-the-loop clinical assessment was done on the SHAP explanations/CNN feature maps nor were any counter-intuitive associations found/reported. The (presumably) overlapping train/val/test data splits during training further complicates the reliability of any post-hoc model interpretation.

4) experiment design: Q: what is the point of comparing your model with Gupta's study? What makes Gupta's study relevant to yours? In general, what is the purpose of the Table 2 comparisons and how do they support the goal of this study?

5) results: (Table 2) Judging by the 0.0 F1 and 0.50 AUC scores, none of these models seem to be trained properly. Are all high AUCs simply due to severe class imbalance in the test set? If so, what was done to protect against/reduce the impact of class imbalance during model training?

6) evaluation scheme:

- (line 118) Q: were the train/val/test data splits done with overlapping subjects? in other words, did the fMRI data slices of one subject end up in both train and test sets?

- Q: Are the classification results reported on test data collected from a seen site (whose data is present in train or val set) or previously unseen ADNI site? Overall, a rigorous evaluation scheme should test the model on an unseen ADNI site, using data of unseen subjects.

7) lack of methodological novelty or technical contributions: all the presented insights on class imbalance and low sample size are well-known in the ML/DL for healthcare community, this study does not present any new approach that can address any of the known/acknowledged issues.

- ## minor issues:

1) manuscript needs significant language and scientific prose/style/structure review:

- one example: ROC plot/performance metrics and interpretability tools must first be introduced in methods section, not in results section!

2) (line 126) missing technical signal preprocessing details

3) should remove all gene-related details from the main text if focus is entirely on fMRI analysis

4) (line 169) can move all NN related details to supplement since these are all standard and well-known.

5) (line 197) Q: what exactly makes the CNN architecture "unified" and not simply a standard CNN? How exactly is adaptation to different input sizes achieved with a fully connected layer (line 183)?

6) (table 1) Q: Are the healthy controls age-matched and sex-matched to the AD/MCI patients? If not, the model may have learnt to predict age or sex instead of disease state.

7) Q: how was data normalization done for this multi-centre dataset, assuming site-specific effects exist and are non-negligible? is this normalization/standardization process used/documented previously by the ML/DL for fMRI community?

- ## misc. questions/comments on interpretability:

1) maybe an interesting ablative experiment: if certain brain regions are known to be associated with AD, does training models after masking those brain regions steeply hurt the model performance or does it learn to use some other feature of the data instead?

2) robustness of interpretation: most model explanation methods have known limitations and robustness issues, i.e., their outputs are not always reliable. There is a serious general issue of author bias when looking at model interpretability results - we overlook the results that we don't expect to see and focus more on what matches our expectations. Some additional experiments are needed to establish robustness of presented insights.

3) model convergence: Q: Do all the compared models use the same set of brain regions when assessing AD risk? or do you get different explanations for each model?

Reviewer #3: The paper proposed an improvement for Alzheimer's Disease (AD) early detection by utilizing 3D Convolutional Neural Network (CNN) to analize fMRI scans. Deep learning has proven effective in many domains, and its application to AD diagnosis represents a novel and powerful approach. The authors employed a substantial dataset and conducted rigorous experiments, demonstrating that 3D CNN-based analysis of fMRI scans achieves high accuracy. Furthermore, their results indicate that gene processing—a commonly used method for AD detection—is less effective than the proposed 3D CNN approach. For future development, I recommend that the authors design a neural network architecture specifically tailored for AD fMRI analysis.

6. PLOS authors have the option to publish the peer review history of their article (what does this mean?). If published, this will include your full peer review and any attached files.

Reviewer #1: No

Reviewer #2: No

Reviewer #3: No

---

## [Author Response · Author response to Decision Letter 0]

22 Sep 2024

Below are our responses to editorial and reviewer comments in plain text, please note that a separate response document (with formatted text and figures) is uploaded and available too.

We sincerely thank the editorial team and the reviewers for providing constructive feedback on our manuscript. In the following section, we will provide responses to the editorial and reviewer comments.

Editors' comments:

 Thank you for providing the style templates, which have been instrumental in refining our manuscript to meet the journal's standards. We have reviewed the templates and guidelines in the links and have made additional changes (i.e. figure 1 -> Fig. 1, section relocation, etc. ) to ensure that our manuscript adheres to these requirements. We have also updated our file naming to ensure they meet the journal’s requirements.

Additionally, the current format of our manuscript was built based on the LaTeX template provided by the journal at https://journals.plos.org/plosone/s/latex. Please let us know if this format is outdated or not applicable. We hope the manuscript now fully complies with PLOS ONE's formatting requirements but are prepared to make any further adjustments if needed.

2. For studies involving third-party data, we encourage authors to share any data specific to their analyses that they can legally distribute. PLOS recognizes, however, that authors may be using third-party data they do not have the rights to share. When third-party data cannot be publicly shared, authors must provide all information necessary for interested researchers to apply to gain access to the data. (https://journals.plos.org/plosone/s/data-availability#loc-acceptable-data-access-restrictions) 

 Thank you for clarification on the requirements for data availability concerning third-party data usage. In response to the guidelines provided, we have added a section to our manuscript titled "Data Availability Statement". This section details the 4 points raised. As cited below:

“Description of the data set and the third-party source: The dataset includes fMRI scans along with demographic information such as age and gender, and clinical data related to Alzheimer’s Disease status (normal control, mild cognitive impairment, or Alzheimer’s Disease). All data were obtained from the Alzheimer’s Disease Neuroimaging Initiative (ADNI).

Verification of permission to use the data set: The use of ADNI data is governed by the ADNI Data Use Agreement, which all researchers must accept before accessing the data. Our study complied with all terms of this agreement. 

Confirmation of whether the authors received any special privileges in accessing the data that other researchers would not have: No special privileges were granted to the authors in accessing the ADNI data. All data used in this study are available to other researchers under the same terms through the ADNI database.

All necessary contact information others would need to apply to gain access to the data: Researchers interested in accessing ADNI data should visit the ADNI website at https://adni.loni.usc.edu/data-samples/adni-data/#AccessData. The website provides detailed instructions on the registration and data access application process.”

We hope this addition adequately addresses the data availability requirements but are ready to modify should there be additional requests.

As requested, we have added the ethics statement in the 'Methods' section of our manuscript, as cited below:

"This study involves secondary analysis of existing public data obtained from the Alzheimer's Disease Neuroimaging Initiative (ADNI). The original collection of the data was approved by the institutional review boards of all participating institutions. Informed consent was obtained from all participants involved in the study. This secondary analysis did not involve direct interaction with human participants, and therefore no additional ethical approval was required for this study."

We want to stress that our study is a secondary analysis of publicly available data, and as such, does not necessitate additional ethical approval. We hope this clarification addresses the concerns and assists in the review process. We appreciate the guidance and the opportunity to clarify this aspect of our research.

4. We note you have included a table to which you do not refer in the text of your manuscript. Please ensure that you refer to Table 3 in your text; if accepted, production will need this reference to link the reader to the Table.

We appreciate the editor’s attention to detail regarding the reference to Table 3. We have included a reference to this table in the supplementary materials section to ensure that it is appropriately linked and accessible. We hope this update meets the journal's requirements. Please let us know if further adjustments are needed.

Reviewers' comments:

Reviewer #1: This manuscript describes a new machine learning algorithm using convolutional networks to detect Alzheimer’s disease from fMRI scan data. The manuscript is well written and the following minor comments and suggestions should be addressed:

1. The authors should consider re-organizing their figure placement such that it appears closer to the section where they are discussed.

We sincerely thank the reviewer for the positive feedback on the clarity of our manuscript and the suggestion regarding the placement of figures. We would like to clarify that the separation of figures and text was due to the submission format guidelines. However, we fully acknowledge the value of the suggestion, we will work with the journal and do our best to reorganize the figures to appear closer to the sections where they are discussed.

2. In the "Deep Learning Frameworks" section, more background information should be given on the baseline RNN and MLP models. The number of trainable parameters in all models should be listed for comparison.

 We thank the reviewer for commenting on the need for additional information. As a response, we have updated the section correspondingly in the new manuscript. For the convenience, we also place it as below:

“Baseline Models

To provide a robust comparison, we utilized several baseline models, each designed to address different aspects of the data characteristics: 

● RNN Model: The baseline Recurrent Neural Network (RNN) model is tailored to capture temporal dependencies within the fMRI data sequences. It utilizes gated recurrent units (GRUs) to address the vanishing gradient problem, making it suitable for learning from time-series data.

● MLP Model: The baseline Multilayer Perceptron (MLP) using a standard fully connected network architecture to process static features extracted from fMRI data.

Trainable Parameters

Here, we provide details on the trainable parameters for each model, demonstrating the computational efficiency of our proposed approach:

● our Model: Approximately 1.3 million (1,331,402) parameters.

● Gupta’s Model: Approximately 2.9 million (2,948,866) parameters.

● RNN Model: Approximately 12.6 million (12,595,586) parameters.

● MLP Model: Approximately 25.2 million (25,174,338) parameters.”

3. Some statistical analysis should be added to see if the performance improvement is statistically significant compared to Gupta's model.

We thank the reviewer for emphasizing the importance of statistically validating the performance improvements of our model compared to Gupta's model. In response, we performed detailed statistical analyses, incorporating both parametric (T-test) and non-parametric (Wilcoxon signed-rank test) methods to assess the significance of the differences observed between the models across multiple datasets.

To provide a thorough evaluation, we analyzed the models' performances on the Resting and PW datasets. The statistical tests revealed that, while the T-test for the Resting dataset suggested only a trend towards significance (T-stat = -1.96, p = 0.0545), the Wilcoxon test indicated a significant difference (W-stat = 311, p = 8.73 * 10-6), indicating a consistent improvement in our model's performance. For the PW dataset, both the T-test (T-stat = -3.07, p = 0.0025) and the Wilcoxon test (W-stat = 6649, p = 2.31 * 10-4) demonstrated statistically significant improvements in model performance. These findings have been documented and discussed in the revised manuscript. We hope that these additions address the concerns raised and strengthen the manuscript. For reviewer’s convenience, we cited the related parts as below:

“We also performed statistical analyses of our and Gupta's model performances on different datasets, as summarized in Table \\ref{tab:model_comparison}. For the Resting test dataset, the T-test (T-stat = -1.96, p = 0.0545) suggested a trend towards a difference in model performance that did not reach conventional statistical significance. However, the significant result from the Wilcoxon test (W-stat = 311, p = $8.73 \\times 10^{-6}$) indicates a consistent difference in the median of the ranked differences, suggesting that one model generally outperformed the other. For the PW dataset, both the T-test (T-stat = -3.07, p = 0.0025) and the Wilcoxon test (W-stat = 6649, p = $2.31 \\times 10^{-4}$) showed significant differences in model performance, indicating consistent discrepancies between the models across this dataset. The significant values from both the T-test and Wilcoxon test for the PW dataset confirm robustness in performance, suggesting that our model is more suited to external conditions represented by this dataset, while performing as well as the Gupta's model.”

4. Did the authors use any data augmentation techniques to compensate the issue of class imbalance? Why or why not? The authors should provide some discussion on this.

We thank the reviewer for bringing up the important issue of class imbalance and the use of data augmentation techniques to address it. In our study, we did not use data augmentation techniques to handle class imbalance, primarily due to concerns about potentially compromising the authenticity and clinical relevance of the complex fMRI data. We were cautious about introducing synthetic variations that might not accurately reflect the underlying biological and medical realities, which are important in the study of Alzheimer’s Disease using fMRI. However, recognizing the importance of this issue, we have added discussion in the "Areas for Improvement" section of our manuscript, as cited below:

“Moreover, addressing class imbalance remains a critical challenge in the effective training of predictive models. In this study, we opted not to use data augmentation techniques to mitigate class imbalance, due to concerns about the authenticity and representation of the complex fMRI data. This decision, while aligned with our goal to preserve data integrity, suggests an area for improvement. Future research could explore the development and validation of sophisticated data augmentation techniques that are capable of enhancing dataset balance without compromising the quality and reliability of the data. This exploration is crucial for improving model performance and ensuring its clinical viability.”

5. Figure 3. The authors should indicate which model was used to generate this plot. If the authors' model was used, it might be worthwhile to compare that with a similar plot generated from Gupta's model for comparison.

We thank the reviewer for the suggestion to clarify which model was used to generate the plots in Figure 3. In response, we have clarified that the plots were generated from our model, while also included plots generated using Gupta's model. To maintain the focus of the main text on our proposed model, we have placed these comparative plots in the supplementary materials. We believe this approach keeps the manuscript streamlined while still providing the detailed comparative data for interested readers. However, we acknowledge the importance of direct comparison and are prepared to incorporate these figures into the main manuscript if the reviewer or editorial board deems it essential for understanding the full context of our study. We are open to making any necessary adjustments to meet the journal's standards and enhance the article's comprehensiveness.

“For further comparative analysis between our proposed model and Gupta's model, we have included additional plots in the supplementary materials.”

Reviewer #2: - ## paper summary:

- This study aims to demonstrate feasibility of using fMRI + standard CNNs to predict risk of AD diagnosis with high performance despite low sample size and highly imbalanced training data. Post-hoc CNN feature map/SHAP explanations suggest that authors' CNN model relies on clinically known brain regions for assessing AD risk.

- ## strengths:

1) publicly available code

2) post-hoc model interpretation seems to align with domain knowledge of AD/MCI

- ## major issues:

1) the scientific goal, focus, or overall argument/hypothesis/structure of the study is not evident

- Q: why is the focus of the study to build a CNN? what purpose is it expected to serve given prior models with AUROCs/accuracies above 0.95/95%?

- the introduction is not thematically aligned with the running title of the paper

We thank the reviewer for the careful analysis and valuable comments, which have provided us with an opportunity to better articulate the aims and significance of our study. Below, we address the points raised:

Our study initially aimed to explore the feasibility of a multimodal approach that incorporates both genetic and fMRI data for predicting Alzheimer's Disease (AD). However, early results indicated that the addition of genetic data did not significantly enhance the predictive performance given the constraints of our dataset—small size and the imbalanced samples. This thus prompted a more focused investigation into an imaging-only model using Convolutional Neural Networks. We optimized the CNN based on this finding and demonstrated it to be more suited for handling the complexities of the data. To further enhance the utility and transparency of the neural network, we integrated interpretability features using techniques such as SHapley Additive exPlanations (SHAP) values, providing insights into potential areas that have higher influence in predicting AD, thereby increasing the clinical relevance and trustworthiness of our predictions.

We chose CNNs due to their superiority in feature extraction from raw imaging data, as well as its less demanding of computing resources, all are points worth considering when dealing with high-dimensional spaces with potentially insufficient computing resources in fMRI studies. CNNs are advantageous for their ability to automate the feature extraction process, reducing the need for extensive preprocessing and making the model more adaptable and easier to implement in clinical settings. This less trainable parameters approach also allows for bet

---

## [Decision Letter · Decision Letter 1]

15 Oct 2024

Deep learning analysis of fMRI data for predicting Alzheimer's Disease: a focus on convolutional neural networks and model interpretability

PONE-D-24-25018R1

Dear Dr. Gerstein,

We’re pleased to inform you that your manuscript has been judged scientifically suitable for publication and will be formally accepted for publication once it meets all outstanding technical requirements.

Kind regards,

Xiaohui Zhang

Academic Editor

PLOS ONE

Additional Editor Comments (optional):

Reviewers' comments:

Reviewer's Responses to Questions

**Comments to the Author**

1. If the authors have adequately addressed your comments raised in a previous round of review and you feel that this manuscript is now acceptable for publication, you may indicate that here to bypass the “Comments to the Author” section, enter your conflict of interest statement in the “Confidential to Editor” section, and submit your "Accept" recommendation.

Reviewer #1: All comments have been addressed

2. Is the manuscript technically sound, and do the data support the conclusions?

Reviewer #1: Yes

3. Has the statistical analysis been performed appropriately and rigorously? 

Reviewer #1: Yes

4. Have the authors made all data underlying the findings in their manuscript fully available?

Reviewer #1: Yes

5. Is the manuscript presented in an intelligible fashion and written in standard English?

Reviewer #1: Yes

6. Review Comments to the Author

Reviewer #1: The authors have sufficiently addressed the reviewer comments, the manuscript can now be considered for publication.

7. PLOS authors have the option to publish the peer review history of their article (what does this mean?). If published, this will include your full peer review and any attached files.

Reviewer #1: No

---

## [Editor Report · Acceptance letter]

24 Oct 2024

PONE-D-24-25018R1 

PLOS ONE

Dear Dr. Gerstein, 

I'm pleased to inform you that your manuscript has been deemed suitable for publication in PLOS ONE. Congratulations! Your manuscript is now being handed over to our production team.

Kind regards, 

on behalf of

Dr. Xiaohui Zhang 

Academic Editor

PLOS ONE